# Would you respect a norm if it sounds foreign? Foreign-accented speech affects decision-making processes

Luca Bazzi[1]* , Susanne Brouwer[2] , Margarita Planelles Almeida[1] , Alice Foucart[1]

**1** Centro de Investigación Nebrija en Cognición, Facultad de Lenguas y Educación, Universidad Nebrija, Madrid, Spain, **2** Centre for Language Studies, Radboud University, Nijmegen, The Netherlands

☯ These authors contributed equally to this work.
* lbazzi@nebrija.es

**Data Availability Statement:** All data files are available from the OSF database (DOI: 10.17605/OSF.IO/2KY3Z).

## Abstract

Does listening to a foreign-accented speaker bias native speakers' behavior? We investigated whether the accent, i.e., a foreign accent versus a native accent, in which a social norm is presented affects native speakers' decision to respect the norm (Experiments 1 and 2) and the judgement for not respecting it (Experiment 2). In Experiment 1, we presented 128 native Spanish speakers with *new* social norms, adapted from the measures imposed by the Spanish Government to fight the Covid-19 pandemic (e.g., 'To avoid the spread of the Covid-19 virus, keep your distance'), whereas in Experiment 2, we presented 240 native Spanish speakers with everyday social norms learned from childhood (e.g., 'Not littering on the street or in public places'), that have an intrinsic cultural and linguistic link. In Experiment 1, the norms were uttered either in a native accent, or in a foreign accent unfamiliar to our participants to avoid stereotypes. In Experiment 2, we added an accent negatively perceived in Spain to assess the role of language attitudes on decision making. Overall, accent did not directly impact participants' final decisions, but it influenced the decision-making process. The factors that seem to underlie this effect are emotionality and language attitudes. These findings add up to the recent Foreign Accent effect observed on moral judgements and further highlight the role of the speaker's identity in decision making.

## Introduction

'Keep your distance' or 'Do not hug your friends or relatives' are two of the multiple new social norms that the COVID-19 pandemic obliged many governments worldwide to impose on the society in order to avoid the spread of the virus. Although they are meant to save lives, how strictly the population respect these norms may depend on several factors, including the identity of the person who states the norm. For instance, our likelihood to respect a norm may change if a doctor, a police officer, or a friend states it, implying that the speaker's identity plays a significant role in one's willingness to respect a norm. One of the indexical properties that reveals the speaker's identity, among others like age or gender, is accent. Accent has been

**Funding:** This study was supported by the Spanish Government (FEDER/Ministerio de Ciencia, Innovación y Universidades – Agencia Estatal de Investigación, FFI2017-83166-C2-2-R, author: LB) and by the Community of Madrid and the European Social Fund (H2019/HUM5772, authors: AF, LB). LB was supported by a grant from the Community of Madrid and European Funds (PEJD-2019-PRE/HUM-16971). The study was realized in the framework of a project funded by the Spanish Government (FEDER/Ministerio de Ciencia, Innovación y Universidades – Agencia Estatal de Investigación, PID2020-115175RB-I00, author: AF). The funders had no role in study design, data collection and analysis, decision to publish, or preparation of the manuscript.

**Competing interests:** The authors have declared that no competing interests exist.

shown to affect the way we process speech [1–5] and to bias our judgments of the speaker [6–12] which have consequences in everyday life. For instance, foreign-accented speakers are less likely to being hired than native speakers [13–16] and they receive harsher sentences than natives in court trials for committing the same crimes [17]. In the present study, we focused on the impact of foreign-accented speech on our social behavior by investigating whether processing norms in a foreign accent affects our decision to respect them.

Although it may appear unlikely at first that a superficial aspect like accent may affect the decisions we make, recent evidence has shown that an a-priori equally superficial aspect—the language we use to process a situation, native versus foreign–modulates our decisions [18, 19] and moral judgements [18, 20–25]. For instance, when presented with the famous Footbridge dilemma [26, 27] in which one must decide whether to kill one person to save five, participants are significantly more likely to perform the action when they read or listen to the dilemma in a foreign language than when they read/listen to it in their native language [18, 26–29]. Important for our purpose, language does not seem to only affect our judgement of moral norms, but also that of social norms. Geipel and colleagues [23] showed that harmful (e.g., 'Sell someone a defective car') and harmless social norms (e.g., 'Fail to keep minor promises') processed in a foreign language provoked a less harsh judgement than those processed in a native language. Hence, using a foreign language reduces sensitivity to both consequences and norms [30]. The main factors that have been proposed to account for this 'Foreign Language effect' (FLe) are a reduction of the emotional response, and an increase of cognitive load and psychological distance in a foreign language compared to a native language [25, 30, 31]. Interestingly, and as we will further describe, these same factors have been reported to be modulated by foreign-accented speech as well [9, 10, 32, 33], which leads to the hypothesis that a foreign accent may also have an impact on native speakers' decisions and behavioral attitudes. The aim of the study is to test this hypothesis.

## Foreign-accented speech and emotion processing

In a recent study, Hatzidaki, Baus, and Costa [32] have reported that the processing of affective words is sensitive to the speaker's accent. They presented native Spanish speakers with neutral, positive and negative words spoken in a native or in a foreign accent. Participants performed a semantic categorization task while their brain activity was recorded using the event-related brain potential (ERP) technique. Behavioral results showed no differences across the two accent conditions. ERP data revealed a larger late positive complex (an ERP component associated with emotional language processing) for emotional words than for neutral words in the two accent conditions, however, although it was true for both positive and negative words in the native accent, the effect was only present for negative words in the foreign accent. The authors concluded that the semantic processing of emotion-laden words is affected by the speaker's accent. They proposed various tentative explanations to account for their observation. First, they considered the possibility that processing of emotional words goes beyond simple word recognition. Based on models of spoken word recognition, they advanced that word recognition not only implies retrieving linguistic information from memory but also extra-linguistic information, like the sounds in which a word was stored. Hence, if the speech signal, such as a foreign accent, does not match the stored properties of the word, recognition may be hampered. Moreover, in the case of emotional words, recognition may also be linked with episodic memory since the emotional aspect of a word may have been encoded in relation with a specific event or situation in a native speech context. Thus, recognition of a word uttered in foreign-accented speech may alter the process. A similar account has been proposed in second language literature, that information or events are more accessible when the

language in which they are referred to is the same as the language in which they were encoded [32–35]. The same reasoning could apply to emotions, which usually provoke a higher response in a native language experienced in natural contexts compared with a foreign language learned in artificial contexts [36–38]. Finally, the authors suggested an additional account that foreign-accented speech may simply be considered as noise [39], which is in line with studies that have shown a reduction of emotion recognition in noisy contexts [40, 41].

## Foreign-accented speech: Cognitive load and psychological distance

Foreign-accented speech involves linguistic alterations compared with more familiar native-accented speech and processing these alterations has been shown to affect sentence comprehension. For instance, native speakers are less sensitive to syntactic errors produced by foreign than native speakers [4, 5], anticipatory mechanisms are reduced in foreign-accented speech [42], stories are remembered in less details in foreign compared to native speech [43] and sensitivity to ironic statements is reduced [44, 45]. Even though a foreign accent may not be an impediment to communication [46], it can provoke a disfluency in language processing, which increases cognitive load [47], and consequently, leads to a negative bias towards the speaker [8, 9, 48]. For instance, Lev-Ari and Keysar [48] presented English native speakers with trivia statements like 'Ants don't sleep' spoken either by native-accented or foreign-accented speakers. Their findings revealed that the processing disfluency provoked by the foreign accents caused foreign speakers to be perceived as less credible, even in the absence of stereotypes that may be associated with the speaker or their culture. The authors proposed that instead of perceiving foreign-accented speakers as more difficult to understand, participants perceived them as less trustful. In line with these conclusions, previous studies have suggested that disfluency may lead to negative language attitudes towards foreign accented speakers, particularly in terms of affect, status and solidarity [1, 7–10]. The negative perception of a foreign-accented speaker is not only due to the disfluency of speech, but also to the social categorization of the speaker. Indeed, by the simple fact of saying 'hello', a speaker reveals her social background and is rapidly categorized as an in- or out-group member [40]. This categorization may trigger stereotypes associated with the speaker's culture [49, 50]. Importantly, the simple fact of being considered a member of a different group generates some psychological distance that may affect how the information coming from the speaker is interpreted [51, 52]. Note that, social categorization, the activation of stereotypes and the effect of disfluency are the three steps that takes place when processing foreign accents. These three steps may occur following an order. According to Mai & Hoffman [50] the social categorization comes first, as literature proved that just by listening a simple word listeners may be able to classify the speaker as a member or their linguistic group or not [53] and this is an automatic activity of the listener. However, the effects of disfluency and associating accent-based stereotypes to the speaker are a more elaborate process that may interfere in the social categorization process when the stereotype effect is sufficient [50]. Literature has not disentangled yet the importance of each one in the foreign processing, even though some exploratory findings have recently been proposed on the importance of social categorization (see [9, 54–56]. Moreover, what has been proposed is that social categorization may be an automatic and spontaneous act that listeners undertake when listening [10]. Literature proved that children may prefer to befriend with native speakers than foreign-speakers [57]. These findings were not associated to the stereotype effect, due to the unlikely possibility that under-five-year-old may have negative stereotypes towards certain types of nationalities. Hence, going back to the main question of the study, if foreign-accented speech, like a foreign language, modulates emotion processing and

increases cognitive load and psychological distance, it may also have an impact on native speakers' decisions and behavioral attitudes. Foucart and Brouwer [33] tested this hypothesis in relation with moral judgements. They presented the Footbridge dilemma described above to Spanish (Experiment 1) and Dutch (Experiment 2) native speakers, uttered either in a native accent or a foreign accent. Results showed an increase in utilitarian decisions in the foreign accent compared to the native accent. The findings suggest that a foreign accent, like a foreign language, is a linguistic context that modulates (neuro)cognitive mechanisms, and consequently, impacts our behavior. Here, we further test the hypothesis of a Foreign Accent effect in relation with social norms. In Experiment 1, participants listened to new social norms adapted from the measures imposed by the Spanish Government to fight the Covid-19 pandemic (e.g., 'To avoid the spread of the Covid-19 virus, you should not hug your friends'), uttered in a native or a foreign accent and indicated how likely they were to respect the norms and how efficient they believed they were. Experiment 2 was similar except that it included everyday social norms learned from childhood, that have an intrinsic cultural and linguistic link (e.g., 'Taking the last seat on a crowded bus').

## General method

As mentioned, in this study we undertook two experiments. In this first part, we include the methodology common to both experiments.

### Participants

In both Experiments 1 and 2, participants were native Spanish speakers from Spain with no hearing problems or disorder. Participants from anywhere in the country could take part, but we ensured the homogeneity of Spanish variety across conditions. Fifteen percent of the overall participants (equally distributed across conditions) spoke one of the coofficial languages of Spain in addition to Spanish. Given that familiarity with other languages may affect accent's perception [5, 58, 59], participants' knowledge of other languages was also controlled for to avoid differences across conditions (all participants undertook a linguistic survey at the end of the experiment in which they were asked about their knowledge in foreign languages. Fifty two percent of participants had an intermediate level, or it was inferior, in a second language and 2% participants lived in a foreign country less than 2.5 years. Participants that spoke a second language were distributed equally across conditions). Both experiments were conducted via online platforms (Experiment 1, SurveyGizmo, www.surveygizmo.com; Experiment 2, Gorilla Experiment Builder, www.gorilla.sc). Participants were recruited via Prolific (www.prolific.co; Experiment 1) and the participant pool of Nebrija Research Centre for Cognition (Experiment 2) and they received the pro-rata of 10 Euros per hour for their participation. The experiments were approved by the Research Ethical Committee of Universidad Nebrija (approval code: UNNE-2020-004) and was conducted in accordance with the Declaration of Helsinki. Participants had to confirm they had received and understood the written information about the consent form to proceed with the experiment.

### Speakers

All audio stimuli in both experiments were recorded by male speakers of similar age, i.e., two native Spanish speakers (Experiment 1: 35 years and Experiment 2: 31 years) for the native accent and an native Indonesian speaker (33 years, Experiments 1 and 2) and an native Arabic speaker from Morocco (35 years, in Experiment 2) for the foreign accents (hereafter, Foreign-Indo and Foreign-Arabic, respectively). Both foreign speakers had an intermediate level of Spanish. Because the social stereotypes associated with an accent and a specific nationality may

affect the perception of the speaker [7, 50], we selected the foreign speakers based on the familiarity of their accent for Spaniards. In Experiment 1, we sought to avoid an effect of stereotype and therefore included an unfamiliar accent, Indonesian. In Experiment 2, we included a familiar accent, usually perceived negatively in Spain, Moroccan. The negative stereotype towards Arabs was confirmed in a recent study that evaluated the attitudes and cultural stereotypes in Spain in which Arabs were rated as the second least liked nationality (22.8% of votes), whereas 'others' corresponded to only 1.5% of the votes [60]. To ensure our participants (did not) recognize the speakers' accent, we asked them to identify them at the end of the experiments. The Moroccan speaker was correctly identified as Arab at 49%, whereas the Indonesian speaker was identified as 'other' at 77% (as well as British English, 11%, American English, 6%, Asian, 6%). The texts to be recorded were given to the speakers who were only asked to read them, hence, recordings in the foreign accent did not include syntactic or semantic errors. For the native accent we had a different speaker in each experiment, due to the unavailability of the speaker of Experiment 1 to record audios for Experiment 2. Both speakers for the native accent had been born and raised in Spain and had a Castilian Spanish standard accent. Speakers had the opportunity to get familiar with the texts before recording them to improve fluency (i.e., avoid pauses, hesitations). All recordings were edited with Audacity software [61] to remove noise and to normalize amplitude and dB levels across audio files.

## Experiment 1: New social norms

In Experiment 1 we tested whether the likelihood to respect *new* social norms (i.e., that emerged as a consequence of the COVID-19 pandemic) is dependent on the accent in which they were uttered. The main variable of interest was the Respect score. However, to verify the consistency of the response, we also asked participants to indicate how efficient they believed the norm was (Efficiency). In addition, to evaluate whether the accent could affect the emotions felt when reading the norms, participants were asked to evaluate their level of anger, disgust, sadness, and fear.

### Participants

An a priori power analysis using the application G*Power [62, 63] using an estimated medium effect size of w = 0.15 and a target power of 0.80 had yielded a minimum requirement of 60 participants. We opted to increase this number to allay any concerns about additional variability introduced from running the study online. One hundred twenty-eight native Spanish speakers (65 females, 62 males, 1 rather not say) took part in the experiment (mean age = 29.49 years; *SD* = 12.13; range = 18–54 years). They were randomly assigned to one of the two accents, Native (*N* = 65, 35 females) and Foreign-Indo (*N* = 63, 30 females).

### Materials

The audio stimuli of Experiment 1 consisted of four new social and behavioral measures established by the Spanish Government to avoid the spread of the Covid-19 (norms were presented orally in Spanish but are translated in English here for convenience):

1. *Due to the threat of a COVID-19 outbreak in Spain, the authorities establish a social distancing of two meters in public places, even between family and friends.*

2. *To avoid the spread of the COVID-19 virus, the use of a mask is mandatory in public spaces, outdoor or indoor, even in informal situations such as birthdays.*

3. *To reduce the transmission of the Coronavirus and to respect the social distancing, any contact, including hugging or kissing should be avoided.*

4. *To contain the Coronavirus pandemic, the regulations advise not to share finger food even among friends and family.*

## Procedure

After filling in a demographic questionnaire, participants were informed they would listen to social and behavioral measures established by the Spanish Government in order to fight the spread of the Covid-19 and would be asked a few questions in relation to these measures. Once they gave their consent, they were randomly assigned to one of the two accents. They were then presented with the blocks Respect and Efficiency (counterbalanced across participants) and then Emotions (this block always appeared after the other two blocks to avoid responses to emotionality biasing the assessment of the respect and efficiency measures). In each block, participants listened to all four new social norms, randomized within the blocks across participants. In the Respect and Efficiency blocks, after listening to each norm, they were asked to answer the question 'How likely are you to respect the measure you listened to?' and 'How efficient do you think this measure is?', respectively, on a sliding scale from 0% (not at all) to 100% (totally). We used this scale to resemble the one used in Lev-Ari and Keysar's [48] study to evaluate the impact of a foreign accent on the veracity of statements. In the Emotion block, participants indicated their feeling after reading each measure based on four basic emotions [64] using the same scale. After completing the three blocks, participants completed a language attitude questionnaire in which they were asked to assess the speaker for linguistic (i.e., '¿How strong was the speaker's accent?', 'How difficult was it to understand the speaker') and social aspects (i.e., 'How trustful/pleasant/educated was the speaker?') on the same scale. Finally, they filled in a linguistic background questionnaire. The experiment lasted for about 12 minutes.

## Data analysis

The data were analyzed using the lm or the cor function from the stats package in R (version 3.2.2) [65]. Three different analyses were conducted. First, to examine whether an accent modulates the likelihood to respect social norms, we conducted a linear regression on the dependent continuous variable Respect (0–100) with Accent as categorical predictor variable. Accent was treatment-coded with the Native accent at the reference level (coded as 0). Secondly, the same analysis was performed with Efficiency (0–100) as dependent variable. Thirdly, to check the consistency of participants' responses for Respect and Efficiency, a Pearson correlation was conducted to analyze the relationship between these two variables.

Emotion has been proven to modulate foreign accent processing and foreign language processing [3, 18, 32, 33], so we ran two additional analyses taking this variable into account. Firstly, we ran a linear regression on each of the emotions as dependent variable (anger, disgust, sadness, and fear; 0–100) with Accent (Native vs. Foreign-Indo) as treatment-coded predictor. Secondly, to explore whether emotion influenced the relationship between Respect and Accent, we ran a multiple regression analysis. The mean of all four emotions was added as a continuous centered predictor into the model. Accent, Emotion and their interaction were directly entered into the model. The same procedure was undertaken to explore whether Emotion influenced the relationship between Efficiency and Accent.

## Results and discussion

Before conducting the main analyses, we first checked whether there was a difference in perception of the native accent and the foreign-Indo accent on a number of linguistic and social

**Table 1. Results of the perception of the native and foreign-Indo accent on a number of linguistic and social variables (0–100).**

| Variables | Native accent | Foreign-Indo accent | t (N = 128) | p | Cohen's d |
|---|---|---|---|---|---|
| Accent strength | 26.49 (26.83) | 79.92 (18.26) | 13.12 | .001 | 2.32 |
| Comprehensibility | 9.07 (15.88) | 25.60 (23.77) | 4.63 | .001 | 0.82 |
| Trustfulness | 43.61 (29.36) | 52.38 (26.29) | 1.77 | .07 | 0.31 |
| Pleasantness | 40.24 (28.61) | 54.57 (28.66) | 2.82 | .005 | 0.50 |
| Educational level | 51.63 (26.20) | 57.73 (28.02) | 1.27 | .20 | 0.22 |

Means (SD) are presented for each accent with the accompanying outcomes of the t-tests.

variables by means of independent *t*-tests (see Table 1). The results demonstrated that the foreign-Indo accent was assessed as significantly stronger than the native accent and harder to understand. The foreign speaker was perceived as significantly more pleasant than the native speaker. No differences were found for trustfulness and educational level.

The analyses revealed no significant effect of Accent on Respect (*estimate* = 1.57, *SE* = 3.35, *t* = 0.34, *p* = .73; $R^2$ = .0009) nor on Efficiency (*estimate* = -4.70, *SE* = 3.35, *t* = -1.4, *p* = .16; $R^2$ = .01). These results suggest that accent was not a significant predictor for the likelihood to respect a norm or for the evaluation of its efficiency. Means are reported in Table 2.

A Pearson correlation showed that Efficiency was strongly related to Respect, *r*(126) = .71, *p* < .001. This positive relationship was present for both the native accent (*r*(63) = .77, *p* = < .001) as well as the foreign-Indo accent (*r*(61) = .67, *p* = < .001).

Mean scores for each emotion across accent conditions are reported in Table 3. Simple regression analyses showed that Accent could not predict any of the emotions (S1 Table in the Supplementary Materials summarizes the results of the models).

Next, a multiple linear regression demonstrated no significant effect of Accent nor a significant interaction between Accent and Emotion on Respect. However, a significant simple effect was found for Emotion, indicating that Respect decreased when the level of Emotion increased for the native accent. Finally, the last analysis demonstrated no significant effect of Accent nor a significant interaction between Accent and Emotion on Efficiency. Results are presented in Table 4.

Experiment 1 did not show an effect of accent on individuals' decisions to respect a norm, that is, participants were equally likely to respect the norms whether they were uttered in a native or a foreign accent. Hence, we failed to observe the Foreign Accent effect reported by Foucart and Brouwer [33] for moral judgments. Note, however, that in their study as well as in studies that investigated the effect of a foreign language on moral judgement, the effect of the native/foreign linguistic context is usually apparent for dilemmas in which the proposed action implies violating a social or moral norms with an emotional component [23]. In Experiment 1, the norms participants had to evaluate were new social norms adapted from the measures imposed by the Spanish Government to fight the Covid-19 pandemic and were presented more as recommendations than well-established social/moral rules. Hence, in Experiment 2, we tested our hypothesis with a different set of norms, learned from childhood to guide our behavior as members of a society. Furthermore, literature on foreign accent has highlighted

**Table 2. Mean scores (SD) for Respect and Efficiency (0–100) for each Accent (native vs. foreign-Indo).**

| | Native accent | Foreign-Indo accent |
|---|---|---|
| Respect | 74.93 (20.43) | 76.09 (17.30) |
| Efficiency | 78.38 (19.30) | 73.68 (18.70) |

**Table 3. Mean scores (SD) for each emotion across Accent (native vs. foreign-Indo).**

|  | **Native accent** | **Foreign-Indo accent** |
|---|---|---|
| Anger | 20.01 (23.11) | 26.63 (28.54) |
| Disgust | 9.04 (17.91) | 12.65 (17.49) |
| Sadness | 42.49 (29.22) | 40.28 (29.03) |
| Fear | 18.49 (24.03) | 20.3 (26.36) |

the role of stereotypes associated to a nationality (or a certain group) that an accent may convey on the perception of the speaker [50, 66]. Thus, to test whether stereotypes could affect individuals' decision to respect a norm, we included an accent that is negatively perceived in Spain (i.e., Moroccan accent).

## Experiment 2: Everyday social norms

While 'Keep your distance' was probably *the* norm of 2020, and listeners were able to rapidly associate it to the COVID-19 pandemic, our lives were already filled with multiple other guidelines or rules. Moral and social norms are part of the society and are based on principals such as welfare (e.g., 'Do not harm other people'), justice (e.g., 'Treat others with respect and fairness'), authority, (e.g., 'Show filial piety'), tradition (e.g., 'Marry before having a baby'), or social norms (e.g., 'Shake hands when you meet someone', [49]). Statements like 'Do not litter on the street' or 'Do not cut in line when in a hurry' may recall our parents, grandparents, teachers or any other authority figure in our childhood days teaching us how to behave in public, while, opposite to "Keep your distance", may let us think of doctors or politics. Moreover, the norms for the COVID pandemic were practically imposed to the society overnight, while moral and social norms have been learnt gradually from childhood, thus, having an intrinsic cultural and linguistic link. Therefore, and most importantly, social norms are associated to the cultural context of learning [67, 68] and may be alterable and context-dependent [49]. Note that in the event of a pandemic, norms imposed to save lives are not context or cultural dependent.

As mentioned in the Introduction, Geipel and colleagues [23] have shown that social norms processed in a foreign language provoked less harsh judgements than those processed in a native language. The authors attempted to justify their results by arguing that the wrongness attitude towards a social or moral transgression may be linked to memory and social-cultural learning process. In other words, they suggested that since social norms are taught early in life

**Table 4. Estimates, standard error, t-values, and p-values of the predictor Accent (native, foreign-Indo), Emotion (mean of anger, disgust, sadness, and fear) and their interaction on Respect and Efficiency.**

|  | **Estimate** | **Std. Error** | **t-value** | **p-value** |
|---|---|---|---|---|
| Respect |  |  |  |  |
| Intercept | 74.40 | 2.27 | 32.67 | .001 |
| Accent | 2.04 | 3.24 | .63 | .53 |
| Emotion | -0.26 | .09 | -2.71 | .007 |
| Emotion by Accent | .09 | .12 | .75 | .45 |
| Efficiency |  |  |  |  |
| Intercept | 78.06 | 2.27 | 34.31 | $< .001$ |
| Accent | -3.85 | 3.24 | -1.18 | 0.23 |
| Emotion | -0.15 | .09 | -1.59 | .11 |
| Emotion by Accent | -0.10 | .12 | -0.78 | .43 |

by relatives or caregivers, they are usually encoded in one's native language, hence, recalling these norms in a native language is easier than doing so in a foreign language. Given that accent is an aspect of language, it is possible that norms encoded in a native accent would be easier to recall if processed in the same native accent than in a foreign accent (see [32] for a similar proposal). To test this hypothesis, we adapted Geipel and colleagues' [23] study.

## Method Experiment 2

In Experiment 2, participants were presented with everyday social norms learned from childhood, that have an intrinsic cultural and linguistic link (e.g., 'No paying for the ticket on public transportation', see S2 Table in the Supplementary Materials) and were asked to indicate 1) how likely they would be to respect a norm, 2) how wrong they evaluated the transgression of a norm, and 3) the level of emotion they felt when listening to the norms. Question 1 and 2 are fairly similar but have a different framing. This manipulation was two-fold: (1) question 1 served to directly compare the respect of new norms (Experiment 1) to everyday social norms (Experiment 2) and (2) question 2 was similar as in Geipel et al. [23] to examine whether a foreign accent would similarly modulate our evaluation as a foreign language does.

## Participants

Results of an a–priori power analysis using G*power [62] showed a minimum of 60 participants (estimated medium effect size of w = 0.15 and a target power of 0.80). As in Experiment 1, we opted to increase the number of participants to allay any concerns about additional variability introduced from running the study online. Two hundred forty native Spanish speakers (180 females, 57 males, 2 rather not say, 3 non-binary) participated (mean age = 28.5, $SD$ = 13.37; range = 18–54 years). They were randomly assigned to one of three accents, native ($N$ = 80, 58 females), foreign-Indo ($N$ = 80, 64 females), or foreign-Arab ($N$ = 80, 58 females).

## Materials

The audio stimuli consisted of twelve everyday social norms spoken in a native or foreign accent (see S2 Table in the Supplementary Materials). The norms were adapted from two different studies [23, 69], originally these norms were used in [70], who selected them from [71], and translated into Spanish by advanced bilinguals when it applied. To select the 12 social norms, we pre-tested a set of 15 norms with 14 native Spanish speakers from Spain who did not take part in Experiment 2 (results are displayed in S1 Table in Supplementary Materials). They were asked to indicate how wrong the transgression of a norm was on a Likert-scale from 0 ('totally fine') to 10 ('totally wrong'). To ensure the transgressions selected for Experiment 2 were sufficiently perceived as wrong, we computed the average scores for each norm (average for all norms = 7.86, $SD$ = 1.51, range = 4.29–9.79) and discarded three norms that had received a low score (i.e., below 7.10).

## Procedure

All participants received written information on the procedure and instructions of the experiment and completed a demographic questionnaire before starting the experiment. Participants listened to social norms and indicated: 1) how likely they were to respect the norm on a sliding scale from 0% ('I always disrespect the norm') to 100% ('I always follow the norm'; Respect), 2) how wrong it was to transgress the norm (0%, 'Extremely wrong'– 100% 'Not wrong at all'; Wrong), and 3) the emotional response elicited when listening to a transgression of a social norm (0%, 'Not at all; 100%, 'Totally'; Emotion). The emotions were the same five used in

Geipel et al.'s [23] study, i.e., anger, disgust, preoccupation, discomfort, and sadness, and were answered on a sliding scale (0%, not at all; 100%, absolutely). Each Respect, Wrong and Emotion score was presented in a separate block that contained four norms (12 in total, norms were counterbalanced across conditions and participants). Based on the score the norms had received for the level of wrongness in the pre-test, they were organized so that the average level was similar in each condition. The experimental session lasted for about 12 min.

## Data analysis

The data were analyzed using the lme4 [72] or the cor function from the stats package in R (version 3.2.2) [65]. Three different analyses were conducted. First, to assess whether accent modulates the likelihood to respect social norms we conducted a linear mixed effect regression [73] on the dependent continuous variable Respect (0–100) with Accent (native, foreign-Indo, foreign-Arab) as categorical predictor. Accent was first treatment-coded with the native accent at the reference level (coded as 0). This created two different contrasts. The first contrast compared participants' decisions on the native accent versus the foreign-Indo accent, while the second contrast compared decisions on the native accent versus the foreign-Arab accent. In addition, Accent was treatment-coded with the foreign-Indo accent at the reference level (coded as 0) such that it was also possible to directly compare the participants' decisions on the foreign-Indo accent versus the foreign-Arab accent. Participants and Items (i.e., type of norms) were added as random effects. The most parsimonious model did not contain any random slopes. Secondly, we conducted the same analyses with Wrong (0–100) as dependent variable. Thirdly, to check the consistency of participants' responses for Respect and Wrong, a Pearson correlation was conducted to analyze the relationship between the two variables.

To assess the influence of Emotion on these effects, we ran two additional analyses. Firstly, we conducted a linear mixed effect regression on each of the emotions as dependent variable (anger, disgust, discomfort, sadness, and preoccupation; 0–100) with Accent (Native (reference level, coded as 0), Foreign-Indo, Foreign-Arab; and Foreign-Indo (reference level coded as 0) vs Foreign-Arab) as treatment-coded predictor. Secondly, to explore whether the emotions influenced the relationship between Accent and Respect, we ran a linear regression. The mean of all emotions was added as a continuous centered predictor into the model. Accent, Emotion and their interaction were directly entered into the model. Finally, the same analysis was performed for Wrong.

## Results

Before conducting the main analyses, we first checked whether there was a difference in perception between the native and the two foreign accents on the linguistic and social aspects by means of a one-way ANOVA (means are reported in Table 5). For the linguistic variables, the results showed a significant effect of Accent on Accent Strength ($F(2,237) = 71.10$, $p < .001$, $\eta_p^2 = .37$). Post hoc tests revealed that compared to the native accent, both the foreign-Indo ($t$

**Table 5. Results of the perception of the native, the foreign-Indo and the foreign-Arab accent on a number of linguistic and social variables (0–100).**

| | Accent strength | Comprehensibility | Trustfulness | Education | Pleasantness |
|---|---|---|---|---|---|
| Native | 32.17 (32.51) | 3.57 (12.24) | 66.91 (21.63) | 72.57 (16.33) | 67.14 (24.38) |
| Foreign-Indo | 74.02 (21.50) | 20.82 (21.87) | 66.40 (20.09) | 73.78 (18.44) | 74.02 (19.41) |
| Foreign-Arab | 75.37 (22.74) | 39.80 (27.69) | 61.20 (21.02) | 58.59 (21.44) | 66.30 (23.48) |

Means (SD) are presented for each accent.

(237) = 10.16, $p < .001$) and the foreign-Arab ($t(237) = 10.48$, $p < .001$) accents' strength were rated stronger. When comparing the two foreign accents, results indicated that there were not significant differences across accents ($t(237) = .32$, $p$ .943). In addition, results showed a significant effect of Accent on Comprehensibility ($F(2,237) = 56.48$, $p < .001$, $\eta_p^2 = .32$). Post hoc tests demonstrated that the foreign-Indo accent ($t(237) = 5.05$, $p < .001$) as well as the foreign-Arab accent ($t(237) = 10.62$, $p < .001$) were less comprehensible than the native accent. Also, results comparing the two foreign accents indicated that the foreign-Indo accent was assessed as significantly less difficult to comprehend than the foreign-Arab ($t(237) = 5.56$, $p < .001$).

For the social variables, results showed a significant effect of Accent on Education ($F(2,237) = 16.05$, $p < .001$, $\eta_p^2 = .11$). Post hoc comparisons demonstrated that the foreign-Arab accented-speaker was rated as less educated than the native accented speaker ($t(237) = -4.69$, $p < .001$), which was not the case when contrasting the native and the foreign-Indo accented-speakers ($p > .1$). Furthermore, when the two foreign accents were taken into account, results indicated that the foreign-Arab was rated as significantly less educated than the foreign-Indo speaker ($t(237) = 5.56$, $p < .001$). Finally, results showed no effects for trustfulness and pleasantness on all contrasts ($p > .1$).

Table 6 presents the results for Respect and Wrong on the three accents. The results showed no effect of Accent on Respect for the first contrast ($\beta_{\text{NativeForeign-Indo}} = 1.61$, $SE = 1.85$, $t = .87$, $p = .38$) nor for the second contrast ($\beta_{\text{NativeForeign-Arab}} = -1.56$, $SE = 1.87$, $t = -.83$, $p = .40$) or the third contrast ($\beta_{\text{Foreign-IndoForeign-Arab}} = -3.17$, $SE = 1.85$, $t = -1.71$, $p = .08$). Analyses also revealed no effect of Accent on Wrong (Contrast 1: $\beta_{\text{NativeForeign-Indo}} = 1.43$, $SE = 2.34$, $t = .61$, $p = .54$; Contrast 2: $\beta_{\text{NativeForeign-Arab}} = -1.90$, $SE = 2.35$, $t = -.81$, $p = .41$; Contrast 3: $\beta_{\text{Foreign-Indo-Foreign-Arab}} = -3.34$, $SE = 2.32$, $t = -1.44$, $p = .15$). In other words, accent was not a predictor for the likelihood to respect an everyday social norm nor for the wrongness level when a moral transgression occurred.

A Pearson correlation showed that Wrong was moderately related to Respect, $r(238) = .28$, $p < .001$. A positive relationship was present for the native Accent ($r(78) = .28$, $p = < .001$) as well as for the foreign-Indo ($r(78) = .28$, $p = < .001$) and the foreign-Arab accent ($r(78) = .15$, $p = .006$).

Linear mixed effect regression models were conducted to examine whether Accent predicted the emotions (see Tables 7 and 8). Results showed an effect of the emotion *sadness* for the first contrast ($\beta_{\text{NativeForeign-Indo}} = -7.55$, $SE = 1.85$, $t = -4.07$, $p < .001$), but not for the second contrast ($\beta_{\text{NativeForeign-Arab}} = -2.19$, $SE = 1.85$, $t = -1.18$, $p = .23$). Similarly, results showed an effect of the emotion *preoccupation* for the first contrast ($\beta_{\text{NativeForeign-Indo}} = 4.91$, $SE = 1.83$, $t = 2.67$, $p = .007$), but not for the second contrast ($\beta_{\text{NativeForeign-Arab}} = -2.35$, $SE = 1.83$, $t = -1.28$, $p = .19$). Furthermore, results showed a significant effect of the emotion *anger* for the first contrast ($\beta_{\text{NativeForeign-Indo}} = 4.65$, $SE = 1.45$, $t = 3.19$, $p = .001$), but not for the second contrast ($p > .1$). Finally, results show an effect of the emotion *discomfort* for both accents (Contrast 1: $\beta_{\text{NativeForeign-Indo}} = 4.99$, $SE = 1.45$, $t = 3.45$, $p = < .001$; Contrast 2: $\beta_{\text{NativeForeign-Arab}} = -3.60$, $SE = 1.45$, $t = -2.47$, $p = .01$).

Table 9 presents the results of the predictors Accent, Emotion and their interaction on Respect. A multiple linear regression demonstrated a significant interaction between Accent

**Table 6. Mean scores (SD) for Respect and Wrong across Accent (native, foreign-Indo, and foreign-Arab).**

|  | Native accent | Foreign-Indo accent | Foreign-Arab accent |
|---|---|---|---|
| Respect | 86.69 (21.33) | 88.69 (21.07) | 86.18 (22.44) |
| Wrong | 81.92 (24.32) | 82.32 (23.25) | 79.27 (24.52) |

**Table 7. Mean scores (SD) for each emotion across accents (native, foreign-Indo, and foreign-Arab).**

|  | Native accent | Foreign-Indo accent | Foreign-Arab accent |
|---|---|---|---|
| Anger | 70.85 (32.99) | 75.51 (29.39) | 68.62 (34.21) |
| Disgust | 49.05 (38.48) | 48.63 (38.02) | 45.38 (38.55) |
| Discomfort | 73.20 (31.38) | 78.20 (27.29) | 69.60 (33.62) |
| Sadness | 47.27 (37.11) | 54.83 (36.27) | 45.08 (37.33) |
| Preoccupation | 56.75 (37.14) | 61.67 (38.30) | 54.40 (38.45) |

and Emotion on Respect for the second contrast ($\beta_{\text{NativeForeign-Arab}}$ = -0.06, $SE$ = .01, $t$ = -3.78, $p$< .001) and the third contrast ($\beta_{\text{Foreign-IndoForeign-Arab}}$ = -0.08, $SE$ = .01, $t$ = 4.72, $p$< .001), but not for the first contrast ($p$> .1). Fig 1 illustrates these interactions: the level of respect increased when the level of emotion increased for the native accent and the Foreign-Indo accent, while the pattern remained stable for the foreign-Arab accent.

In addition, results demonstrated a significant simple effect of Accent for the first contrast ($\beta_{\text{NativeForeign-Indo}}$ = 1.66, $SE$ = .45, $t$ = 3.63, $p$< .001) and the third contrast ($\beta_{\text{Foreign-IndovsArab}}$ = -2.22, $SE$ = .46, $t$ = -4.83, $p$< .001), indicating that the level of respect was higher for the Foreign-Indo accent than for the native accent or the Moroccan accent when emotion was held constant at the mean.

Finally, results showed a simple effect of Emotion for the native accent ($\beta$ = .06, $SE$ = .01, $t$ = 4.73, $p$< .001) and for the Foreign-Indo accent ($\beta$ = .07, $SE$ = .01, $t$ = 5.84, $p$< .001), indicating that when the level of emotion increased the level of Respect also increased.

Subsequently, we conducted additional analyses to examine whether the perception participants had about the accents could contribute to the effect of Accent on Respect. Hence, we included the significant linguistic and social variables, i.e., Accent Strength,

**Table 8. Estimates, standard error, t-values, and p-values of the predictor Accent (Accent$_{\text{NativeForeign-Indo}}$ and Accent$_{\text{NativeForeign-Arab}}$) on the five emotions (anger, disgust, discomfort, sadness, and preoccupation).**

|  | Estimate | Std. Error | t-value | p-value |
|---|---|---|---|---|
| **Anger** |  |  |  |  |
| Intercept | 70.85 | 1.02 | 68.87 | .001 |
| Accent$_{\text{NativeForeign-Indo}}$ | 4.65 | 1.45 | 3.19 | .001 |
| Accent$_{\text{NativeForeign-Arab}}$ | -2.23 | 1.45 | -1.53 | .12 |
| **Disgust** |  |  |  |  |
| Intercept | 49.05 | 1.36 | 35.97 | .001 |
| Accent$_{\text{NativeForeign-Indo}}$ | -.42 | 1.92 | -0.22 | .82 |
| Accent$_{\text{NativeForeign-Arab}}$ | -3.66 | 1.92 | -1.90 | .057 |
| **Discomfort** |  |  |  |  |
| Intercept | 73.20 | 1.02 | 71.17 | .001 |
| Accent$_{\text{NativeForeign-Indo}}$ | 4.99 | 1.45 | 3.45 | < .001 |
| Accent$_{\text{NativeForeign-Arab}}$ | -3.60 | 1.45 | -2.47 | .01 |
| **Sadness** |  |  |  |  |
| Intercept | 47.27 | 1.31 | 36.05 | .001 |
| Accent$_{\text{NativeForeign-Indo}}$ | 7.55 | 1.85 | 4.07 | < .001 |
| Accent$_{\text{NativeForeign-Arab}}$ | -2.19 | 1.85 | -1.18 | .23 |
| **Preoccupation** |  |  |  |  |
| Intercept | 56.75 | 1.29 | 43.76 | .001 |
| Accent$_{\text{NativeForeign-Indo}}$ | 4.91 | 1.83 | 2.67 | .007 |
| Accent$_{\text{NativeForeign-Arab}}$ | -2.35 | 1.83 | -1.28 | .19 |

**Table 9. Estimates, standard error, t-values, and p-values of the predictors Accent (Accent$_{NativeForeign-Indo}$, Accent$_{NativeForeign-Arab}$ and Accent$_{Foreign-IndoForeign-Arab}$), Emotion (mean for anger, disgust, discomfort, sadness, and preoccupation), and their interaction on Respect for the first, second (top panel), and third contrast (bottom panel).**

| First and second contrast | Estimate | Std. Error | t-value | p-value |
|---|---|---|---|---|
| Intercept | 86.72 | .32 | 269.06 | < .001 |
| Accent$_{NativeForeign-Indo}$ | 1.66 | .45 | 3.63 | < .001 |
| Accent$_{NativeForeign-Arab}$ | -0.56 | .45 | -1.22 | .22 |
| Emotion | .06 | .01 | 4.73 | < .001 |
| Emotion by Accent$_{NativeForeign-Indo}$ | -0.01 | .01 | 1.48 | .29 |
| Emotion by Accent$_{NativeForeign-Arab}$ | -0.06 | .01 | -3.78 | < .001 |
| Third contrast | | | | |
| Intercept | 88.39 | .32 | 270.73 | < .001 |
| Accent$_{Foreign-IndoForeign-Arab}$ | -2.22 | .46 | -4.83 | < .001 |
| Emotion | .07 | .01 | 5.84 | < .001 |
| Emotion by Accent$_{Foreign-IndoForeign-Arab}$ | -0.08 | .01 | -4.72 | < .001 |

Comprehensibility, and Education, in the model in interaction with Accent. None of these variables were significant (see S3 Table in the Supplementary Materials).

Finally, Table 10 presents the results of the predictors Accent, Emotion and their interaction on Wrong. A multiple linear regression demonstrated a significant interaction between Accent and Emotion on Wrong for the first contrast ($\beta_{NativeForeign-Indo}$ = .05, $SE$ = .02, $t$ = 2.22, $p$ = .02), but not for the second contrast (native vs. foreign-Arab, $p > $ .1) nor for the third contrast (foreign-Indo- vs. foreign-Arab; $p > $ .1). Fig 2 shows that the wrongness level is higher for the native accent when emotion is low, whereas it is higher for the foreign-Indo accent when emotion is high.

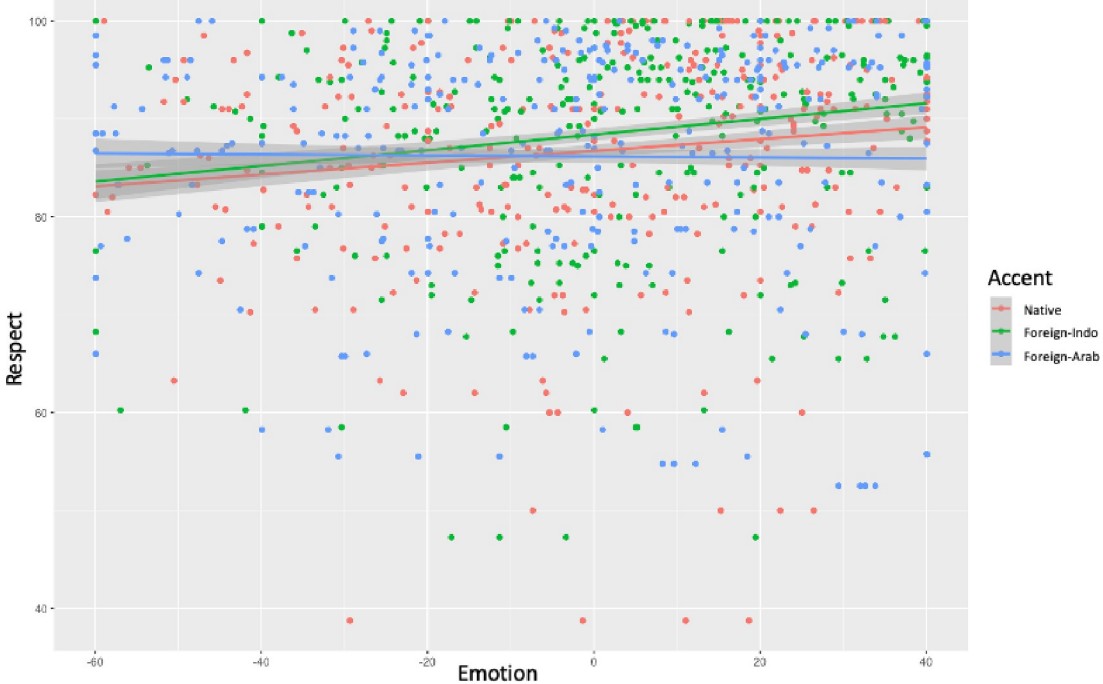

**Fig 1. Interaction between Accent (Native Accent vs. Foreign-Indo vs. Foreign-Arab) and Emotion (mean for anger, disgust, discomfort, sadness, and preoccupation centered) on Respect.**

**Table 10. Estimates, standard error, t-values, and p-values of the predictors Accent (Accent$_{NativeForeign-Indo}$, Accent$_{NativeForeign-Arab}$, and Accent$_{Foreign-IndoForeign-Arab}$), Emotion (mean for anger, disgust, discomfort, sadness, and preoccupation), and their interaction on Wrong for the first, second (top panel), and third contrast (bottom panel).**

| First and second contrast | Estimate | Std. Error | t-value | p-value |
|---|---|---|---|---|
| Intercept | 81.97 | .40 | 200.77 | < .001 |
| Accent$_{NativeForeign-Indo}$ | -0.22 | .58 | -0.39 | .69 |
| Accent$_{NativeForeign-Arab}$ | -2.27 | .57 | -3.92 | < .001 |
| Emotion | .09 | .01 | 6.02 | < .001 |
| Emotion by Accent$_{NativeForeign-Indo}$ | .05 | .02 | 2.22 | .02 |
| Emotion by Accent$_{NativeForeign-Arab}$ | .03 | .02 | 1.40 | .15 |
| Third contrast | | | | |
| Intercept | 81.75 | .41 | 197.66 | < .001 |
| Accent$_{Foreign-IndoForeign-Arab}$ | -2.04 | .58 | -3.51 | < .001 |
| Emotion | .14 | .01 | 8.64 | < .001 |
| Emotion by Accent$_{Foreign-IndoForeign Arab}$ | -0.02 | .02 | -0.94 | .34 |

Furthermore, results demonstrated a simple effect of Accent for the second contrast ($\beta_{Native-Foreign-Arab}$ = -2.27, $SE$ = .57, $t$ = -3.92, $p$< .001), indicating that the level of wrong is higher for the native accent than for the foreign-Arab accent when emotion was held constant at the mean. A similar simple effect of Accent was found for the third contrast ($\beta_{Foreign-IndoForeign-Arab}$ = -2.04, $SE$ = .58, $t$ = -3.51, $p$< .001), indicating that the level of wrong is higher for the foreign-Indo accent than for the foreign-Arab accent when emotion was held constant at the mean.

Finally, results showed a simple effect of Emotion for the native accent ($\beta$ = .09, $SE$ = .01, $t$ = 6.02, $p$< .001) and for the foreign-Indo accent ($\beta$ = .14, $SE$ = .01, $t$ = 8.63, $p$< .001). In both

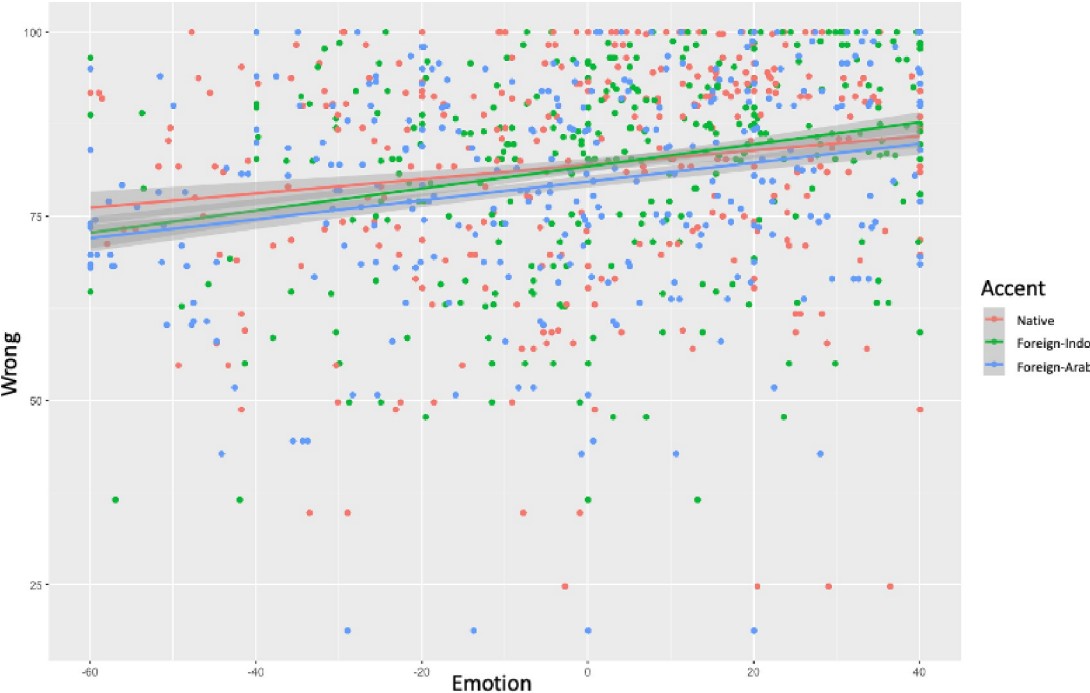

**Fig 2. Interaction between Accent (Native Accent vs. Foreign-Indo vs. Foreign-Arab) and Emotion (mean for anger, disgust, discomfort, sadness, and preoccupation centered) on Wrong.**

cases, this indicates that when the level of emotion increased the level of wrongness also increased.

In Experiment 2, we did not observe a direct effect of accent on the likelihood to respect a norm or on the evaluation of the wrongness of its transgression (we only report a simple effect of Accent when emotion was held constant at the mean). However, we found that the emotions triggered by the norms modulated the decision and that this modulation was affected by accent. Furthermore, accent significantly affected emotional response, but this effect varied according to the speaker. Variables such as social categorization, stereotypes, the difficulty in comprehend the speech or acoustic differences might have played a role in the results. The implication of these results is further considered in the Discussion.

## Cross-experiment analyses

To confirm our claim that individuals are more sensitive to everyday social norms learned from childhood (Experiment 2) than to new social norms imposed due to the Covid-19 pandemic (Experiment 1), we compared the results for Respect and Emotion across experiments in an independent t-test. As mentioned earlier, social norms have an intrinsic cultural and linguistic link and they are associated to the cultural context of learning, hence, we checked our manipulation for the native accent only because the results for the foreign accents could be biased by a different learning context. As we expected, participants were significantly more likely to respect everyday social norms than new social norms ($t(143)$ = -4.33, $p < .001$, $d$ = 0.70), and they were also more emotionally responsive to them ($t(143)$ = -13.49, $p < .001$, $d$ = 2.23).

## Discussion

The current study aimed to investigate whether foreign-accented speech has an impact on native speakers' decisions and behavioral attitudes. We analyzed the potential effect of a foreign accent when processing social norms across two experiments. In Experiment 1, we presented participants with *new* social norms, adapted from the measures imposed by the Spanish Government to fight the Covid-19 pandemic, whereas in Experiment 2, we presented them with *everyday* social norms learned from childhood, that have an intrinsic cultural and linguistic link. In both experiments, the norms were uttered either in a native accent or in a foreign accent. In Experiment 1, we included a foreign accent that was non-familiar to Spaniards to avoid stereotypes associated with a specific nationality. In Experiment 2, in contrast, we added an accent which is usually negatively perceived in Spain to test the possibility that the Foreign Accent effect may be driven by stereotype [33]. We were particularly interested in the responses to the likelihood to respect the social norms depending on the accent in which they were uttered.

Results in both experiments revealed that the level of likelihood to respect new and everyday social norms was independent of the accent. A similar result was found regarding the level of wrongness of the transgression of an everyday social norm (Experiment 2). However, we found that both Respect and Wrongness were significantly modulated by the emotions triggered by the norms, and this modulation was affected by accent. Note that this modulation was observed only when contrasting native vs. foreign-Indo and foreign-Indo vs. foreign-Arab accents, which suggests that the impact of accent on our behavior may vary across accents, as previously reported (Foucart & Brouwer [33]). This result was found in response to norms to which participants were more sensitive (i.e., everyday social norms). Hence, even though foreign-accented speech did not directly impact participants' final decisions, it did influence the decision-making process. We interpret the results in view of our original hypothesis.

One of the factors that originally made us think that a foreign accent may modulate our decisions is emotion processing. Recall that judgements for the transgression of moral/social norms have been reported to be less harsh in a foreign *language* than in a native *language* [23, 24], mainly due to a reduction of the emotional response in a foreign compared to a native language [38, 74]. Regarding emotionality and foreign *accent*, Hatzidaki and colleagues' [32] electrophysiological results indicated that foreign-accented speech had an impact on semantic processing of emotional (positive) words, leading to lesser sensitivity compared to a native accent. In contrast, the behavioral results of their semantic categorization task revealed no differences based on the speaker's accent.

In our study, we assessed the effect of foreign accent on emotionality by evaluating the degree of emotion felt by participants after listening to moral-social norms. We observed that, in comparison with the native accent, participants felt more anger, discomfort, sadness, and preoccupation when norms were uttered by the Indonesian speaker but felt less discomfort when they were uttered by the Moroccan speaker. Hence, foreign-accented speech seems to indeed modulate emotionality, however, in contrast with our prediction, it does not always lead to a reduction. Note that this modulation of emotionality by accent was significant only in Experiment 2, which involved norms to which participants were more sensitive than to norms used in Experiment 1. This is consistent with the observation that the linguistic context–language or accent—affects mainly decisions with an emotional component [18, 23, 24].

Interestingly, the emotions felt by the participants varied according to the speakers. The unfamiliar accent (Indonesian) provoked a more intense emotional response than the native accent, whereas the negative accent (Moroccan) provoked a less intense one comparing to the native accent. Recall that the Moroccan accent was assessed as more difficult to comprehend than the Indonesian accent. We hypothesized that the difficulty in the comprehension process may have affected differently the elicitation of emotions due to the cognitive effort associated when processing the Moroccan accent. Comprehensibility and Accent Strength were reflected in the way emotion modulated participants' decisions. Indeed, although we found that the more intense the emotional response, the higher the level of wrongness for the three accents, this effect was significantly larger for the Indonesian accent. In contrast, while the level of respect was similarly modulated by emotion in the native accent and Indonesian accents, it was not so in the Moroccan accent. Hence, accent seems to be a linguistic aspect that modulates emotionality. Further research is need to shed light on the effects of Accent strength and Comprehensibility on the emotion processing when listening to foreign accents. The bias of accent on the emotions might have affected the decision-making process in our experiment, however, this modulation may be driven by the language attitudes related to the speaker, which leads us to our next points.

Another factor that led us to believe that a foreign accent may modulate our decisions is the processing disfluency generated by the difficulty to understand the speaker, which increases cognitive load. In our study, although participants rated the comprehensibility harder for the foreign speakers than for the native speaker, results indicated that this variable did not influence participants' decisions, even if the Foreign-Arab was significantly assessed more difficult to comprehend than the Foreign-Indo. This result contrasts with Foucart and Brouwer's [33] study that suggested an implication of comprehensibility in the Foreign Accent effect they reported on moral judgements. Note, however, that their study involved various foreign accents and various speakers and that significant differences were not observed with all of them. It is possible that the foreign accents we used were not strong enough to generate a disfluency that would modulate our participants' decisions. This explanation is consistent with Lev-Ari and Keysar's [48] observation that the bias in participants' decision to believe foreign speakers' statements was modulated by the strength of their accent. More research is needed to evaluate how different degrees of disfluency affect decision making.

Disfluency may not only increase cognitive load, it has also been suggested to lead to negative language attitudes towards foreign-accented speakers [1, 8–10]. Here, the two accents we used revealed different attitudes. As expected, the speaker with the Moroccan accent, which is associated with negative stereotypes [60], was perceived as significantly less educated than the native speaker. Even compared to the Indonesian accent, the Moroccan accent was evaluated significantly less educated. This is consistent with the tendency to attribute lower status to speakers identified with specific nationalities that are negatively perceived by native speakers [7]. In contrast, the speaker with the Indonesian accent, an unfamiliar accent to Spaniards who was not associated to specific stereotypes, was assessed as significantly more pleasant than the native speaker in both experiments. Although there was no significant direct impact of accent on Respect or Wrongness, our findings revealed that these two accents had an impact on emotionality and on the decision-making processes, but they affected them in different ways. Particularly interesting is the fact that the Indonesian speaker was perceived the most pleasant speaker, thus it can be hypothesized that this might have affected positively the results. Nevertheless, this impression of the foreign-accented speaker did not affect the results positively; the likelihood for respecting the norms did not increase nor the wrongness level associated to the disrespect of norms. On the contrary, even if the Moroccan speaker was perceived as less educated, the effect of this variable did not seem to have affected the responses of the native listeners (we thank the reviewer for these comments). However, results suggests that language attitudes and the psychological distance generated by an accent do influence cognitive process and emotionality. Note, however, that recent studies have proposed exploratory findings suggesting that processing foreign-accented speech is more related to the acoustic differences of the speech (e.g., the length of the stimuli) rather than the identity of the speaker *per se*. [75]. This supports the idea that materials included in the experiments should be either as equal as possible across conditions to reinforce the validity of the results or materials that should control the intelligibility of the speech across accent conditions (e.g., using images to associate to speakers and not audios stimuli, see [9, 54, 55]. Future research needs to address this issue in order to better identify whether acoustic differences across stimuli or to which degree some aspects related to the speaker's identity (e.g., stereotyping) affect the decision-making process.

In conclusion, the key objective of the investigation was to investigate the impact of the speaker's identity on decision making, more specifically, the impact of a foreign accent on the likelihood to respect social norms. Based on the literature, we hypothesized that since foreign-accented speech reduces emotion processing and increases cognitive load and psychological distance, it may modulate native speakers' decisions, similarly as what has been observed with the use of a foreign language [18, 19]. Our results revealed that accent did not directly impact participants' final decisions, but it influenced the decision-making process. The factors that seem to underlie this effect of accent are emotionality and language attitudes related to the speaker. These findings add up to the recent Foreign Accent effect observed on moral judgements [33] and further highlight the role of the speaker's identity in decision making. Further research is needed to better understand the consequences the interaction between a native and a foreign speaker may have on our behavior, which is essential in our multilingual world.

## Supporting information

**S1 Table. Estimates, standard error, t-values, and p-values of the predictor Accent (native, foreign) on the four emotions (anger, disgust, sadness, and fear).**
(DOCX)

**S2 Table. English translation of the everyday social norms included in Task 1 of Experiment 2 (norms marked with '\*' were not included because of a low rating in the pre-test).** Means (SDs) refer to the rating attributed to the norms in the pre-test. The structure of the sentences was adapted to match the Spanish translation of the questions 'How likely are you to respect the norm. . .' (Respect variable) and 'How wrong is it to. . .' (Wrong social variable). (DOCX)

**S3 Table. Estimates, standard error, t-values, and p-values of the predictor Accent (native, foreign-Indo, foreign-Arab) in interaction with Accent Strength, Comprehensibility and Education on Respect.**
(DOCX)

## Author Contributions

**Conceptualization:** Luca Bazzi, Susanne Brouwer, Margarita Planelles Almeida, Alice Foucart.

**Formal analysis:** Luca Bazzi, Susanne Brouwer.

**Funding acquisition:** Alice Foucart.

**Investigation:** Luca Bazzi, Susanne Brouwer, Margarita Planelles Almeida, Alice Foucart.

**Methodology:** Luca Bazzi, Susanne Brouwer, Alice Foucart.

**Writing – original draft:** Luca Bazzi, Susanne Brouwer, Alice Foucart.

**Writing – review & editing:** Luca Bazzi, Susanne Brouwer, Margarita Planelles Almeida, Alice Foucart.

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
