## [Decision Letter · Decision Letter 0]

28 Apr 2022

PONE-D-22-08900Would you respect a norm if it sounds foreign? Foreign-accented speech affects decision-making processes.PLOS ONE

Dear Dr. Foucart,

Thank you for submitting your manuscript to PLOS ONE. I was able to get advice from one expert reviewer. As you will see, the reviewer was positive about your manuscript, but also made detailed comments on some aspects of your study, mainly regarding the justification of the studies and the lack of clarity of some of the analyses. We invite you to submit a revised version of the manuscript that addresses the points raised during the review process.

We look forward to receiving your revised manuscript.

Kind regards,

José A Hinojosa, Ph.D.

Academic Editor

PLOS ONE

Journal Requirements:

2. Peer review at PLOS ONE is not double-blinded (https://journals.plos.org/plosone/s/editorial-and-peer-review-process). For this reason, authors should include in the revised manuscript all the information removed for blind review.

3.We note that the grant information you provided in the ‘Funding Information’ and ‘Financial Disclosure’ sections do not match. 

"This study was supported by the Spanish Government (FEDER/Ministerio de Ciencia, Innovación y Universidades – Agencia Estatal de Investigación, FFI2017-83166-C2-2-R, author: LB) and by the Community of Madrid and the European Social Fund (H2019/HUM5772, authors: AF, LB). LB was supported by a grant from the Community of Madrid and European Funds (PEJD-2019-PRE/HUM-16971). The study was realized in the framework of a project funded by the Spanish Government (FEDER/Ministerio de Ciencia, Innovación y Universidades – Agencia Estatal de Investigación, PID2020-115175RB-I00, author: AF)."

Reviewers' comments:

Reviewer's Responses to Questions

**Comments to the Author**

1. Is the manuscript technically sound, and do the data support the conclusions?

Reviewer #1: Partly

2. Has the statistical analysis been performed appropriately and rigorously? 

Reviewer #1: No

3. Have the authors made all data underlying the findings in their manuscript fully available?

Reviewer #1: Yes

4. Is the manuscript presented in an intelligible fashion and written in standard English?

Reviewer #1: Yes

5. Review Comments to the Author

Reviewer #1: In this manuscript, the authors address a very interesting and socially relevant topic: does the accent in which we listen to social norms influence our behavior? Moreover, they have a large sample size, which is commendable. However, there are some theoretical and methodological issues that prevent me from recommending this article for publication in this state. Please see more detailed comments below:

There are two main points in the manuscript that seem to me to be poorly justified:

On the one hand, the second study is justified in part because more established social norms are supposedly acquired in interactions with native speakers and are more emotional. But I don't see why social norms related to COVID would not be acquired in interactions with native speakers or would be less emotional.

On the other hand, the other justification for the second study is based on the idea that participants would be more familiar with the Arabic accent, and this might lead to more negative biases. However, neither of these two issues ends up being properly demonstrated (less than half of the participants correctly recognized the accent of the speaker; nor did they have more negative biases when listening to this speaker).

Hence, I would recommend the authors to try to justify the second study with other, clearer premises.

At some points, authors do not follow the journal’s referencing style. Please review it.

Abstract, lines 30-31: “or in a foreign accent unfamiliar to our participants to avoid stereotypes”. How do you know that the unfamiliar accent did not activate stereotypes towards foreign-accented speakers (outgroup members)?

As you write in the Introduction (lines 121-124): “The negative perception of a foreign-accented speaker is not only due to the disfluency of speech, but also to the social categorization of the speaker. Indeed, by the simple fact of saying ‘hello’, a speaker reveals her social background and is rapidly categorized as an in- or out-group member”.

This is an issue that may be addressed throughout the manuscript.

Sub-section “Foreign-accented speech and emotion processing”, lines 88-95. I would recommend the authors to check a recently published paper in which this perspective on the effects of foreign-accented speech on lexical-semantic access is discussed:

Romero-Rivas, C., & Costa, A. (2022). On the flexibility of the sound-to-meaning mapping when listening to native and foreign-accented speech. Cortex, 149, 1-15.

According to this paper, and following more recent models of speech processing (e.g., Hickok & Poeppel, 2007; Gwilliams, Poeppel, et al., 2018), non-standard speech signals (such as the ones produced by foreign-accented speakers) do not necessarily impair access to lexical-semantic information. Therefore, the effects obtained by Hatzidaki et al. (2015) (and by other authors in other studies) may be alternatively explained by how the acoustic properties of the native vs. foreign-accented words (e.g., longer duration of the words in the latter vs. former) might artificially affect the ERPs.

I’d just like to suggest that one should consider the possibility that the effects found by these authors could be explained by the properties of the acoustic signal (such as the duration of the stimuli), and not so much by the identity of the speakers.

This comment does not only apply to Hatzidaki et al., but also to the papers that are reviewed below in the next sections.

Participants sub-section, lines 155-157: “Given that familiarity with other languages may affect accent’s perception (5,55,56), participants’ knowledge of other languages was also controlled for to avoid differences across conditions”.

Could you please clarify what you meant here, and how did you control for this variability across participants?

Speakers sub-section, lines 174-175: “In Experiment 1, we sought to avoid an effect of stereotype and therefore included an unfamiliar accent, Indonesian”.

It is not clear to me whether including an, a priori, unfamiliar accent, would prevent participants for activating stereotypes. For instance, an unfamiliar accent may activate outgroup membership associations, and potentially generate negative biases towards that person.

Speakers sub-section, lines 175-176: “In Experiment 2, we included a familiar accent, usually perceived negatively in Spain, Moroccan”.

The problem is that, later in the same paragraph, you mention that only 49% of participants were able to correctly recognize the speaker’s accent. Thus, any conclusion about the activation of stereotypes specifically associated with this accent is difficult to sustain.

Lines 250-252: Isn’t it redundant to say twice that the Native accent condition was dummy-coded as 0?

Lines 297-300: Why didn’t you carry out the same analysis on Efficiency?

Line 389: It is not clear what you meant by “Task 1”. Did participants complete more than one task?

Lines 419-424: You should also report whether there were any differences between the foreign-Indo and foreign-Arab accents in terms of Accent Strength and Comprehensibility.

Same applies to Education (lines 427-429).

Lines 435-441: Before reporting the individual comparisons between specific conditions, you should carry out/report the analysis for the main factor Accent (both when analyzing Respect and Wrong). This can be done by dummy coding the three levels of the main factor (e.g., Native Accent = 0; Foreign-Indo = 1; Foreign-Arab = 2; since you were expecting larger effects in the latter condition).

Same applies to the analyses on Emotion. These analyses could be either linear mixed models or ANOVAs.

Lines 470-478: I am sorry, but I am not able to understand this analysis. I believe that you entered each potential comparison between the native and the foreign accents as predictors in the model; then, emotion and the interactions between emotion and the comparisons were also entered into the model. Why did you do this? Is there any particular reason to justify this decision?

Possibly, a more appropriate way to conduct this analysis would be to include the main effect of accent, the main effect of emotion, and the interaction between the two factors. Then, if you find a main effect of accent, you may proceed and check the post-hoc comparisons. Same would happen with the interaction between factors. Also, at no point you explore the potential difference between the two foreign accents, and that may be informative as well.

Same applies to the analysis reported in lines 491-498. These analyses could be either linear mixed models or ANOVAs.

Also, if the interaction between Accent and Emotion ends up being significant, I think that readers would appreciate a graphic representation of this interactions, so they can properly interpret/follow the results.

Lines 513-517: These conclusions (and the ones in the next section) may change after carrying out the analyses I suggest above.

Lines 574-578: If I am not wrong, you did not include participants’ fluency ratings in your regression models. Therefore, I do not understand why you say that “this variable did not influence participants’ decisions”.

Lines 588-598: These conclusions are interesting, but my feeling is that the results went in the opposite direction of your hypotheses, didn’t they? If the Indonesian speaker was perceived as more pleasant, one may expect to observe positive effects on decision-making; and the opposite may be expected from the Moroccan speaker, who was perceived as less educated. However, the pattern of results is contradictory with the expectations. Have you thought about any explanation for this pattern?

6. PLOS authors have the option to publish the peer review history of their article (what does this mean?). If published, this will include your full peer review and any attached files.

Reviewer #1: No

---

## [Author Response · Author response to Decision Letter 0]

27 May 2022

Please, see the 'Response to Reviewers' document attached.

---

## [Decision Letter · Decision Letter 1]

30 Jun 2022

PONE-D-22-08900R1Would you respect a norm if it sounds foreign? Foreign-accented speech affects decision-making processes.PLOS ONE

Dear Dr. Foucart,

Thank you for submitting your manuscript to PLOS ONE. One original reviewer submitted comments to your revised manuscript. While s/he feels that your manuscript is now much improved, the reviewer also noted some remaining corcerns, mainly related to the output and interpretation of some statistical analyses.

We look forward to receiving your revised manuscript.

Kind regards,

José A Hinojosa, Ph.D.

Academic Editor

PLOS ONE

Reviewers' comments:

Reviewer's Responses to Questions

**Comments to the Author**

1. If the authors have adequately addressed your comments raised in a previous round of review and you feel that this manuscript is now acceptable for publication, you may indicate that here to bypass the “Comments to the Author” section, enter your conflict of interest statement in the “Confidential to Editor” section, and submit your "Accept" recommendation.

Reviewer #1: (No Response)

2. Is the manuscript technically sound, and do the data support the conclusions?

Reviewer #1: Partly

3. Has the statistical analysis been performed appropriately and rigorously? 

Reviewer #1: I Don't Know

4. Have the authors made all data underlying the findings in their manuscript fully available?

Reviewer #1: Yes

5. Is the manuscript presented in an intelligible fashion and written in standard English?

Reviewer #1: Yes

6. Review Comments to the Author

Reviewer #1: I would like to congratulate the authors for the revision of their manuscript. I believe that they present a more robust version this time, but I still found some relevant issues that should be addressed before the article can be published. Please find more detailed comments below:

- I still detected some inconsistencies in the referencing style (e.g., lines 80, 120, 145-146). Please recheck the document to make sure you are following the journal's referencing style.

- Line 165, typo: it should read “Experiments” rather than “Experiment”.

- Lines 344-347: Based on this comment of the authors (i.e., that stereotypes associated with a group might affect the decision to accept a norm; which is repeated in the Discussion, lines 578-579), I still consider that it would be important to compare the two foreign accents: if indeed the accent effect found by the authors is due to the stereotype generated by a recognizable group (i.e., Moroccan), the expected result would be that the Moroccan accent produces a greater modulation of respect for the norm than the Indonesian accent. This could be addressed by using the Accent factor (with three levels: native, foreign-Moroccan, foreign-Indonesian) as a fixed factor in the analysis, or by adding the foreign-Moroccan vs. foreign-Indonesian accent comparison to the model. You have the data available, so it would be a shame not to perform this analysis.

- Line 369: “(…) they suggested that since social norms are learned early in life by relatives or caregivers…”. It should read: “(…) they suggested that since social norms are TAUGHT early in life by relatives or caregivers…”.

- Line 468: “foreign-Arab was significantly less educated than the foreign-Indo speaker”. It should read: “foreign-Arab was RATED AS significantly less educated than the foreign-Indo speaker”.

- Figure 1: My apologies, but I do not understand this figure. In the text you state: “a multiple linear regression demonstrated a significant interaction between Accent and Emotion on Respect for the second contrast (βNativevsForeign-Arab= -0.06, SE= .01, t= -3.78, p< .001; Fig 1), but not for the first contrast (p> .1). This indicates that the level of respect increased when the level of emotion increased for the native accent, while the pattern remained stable for the foreign-Arab accent”. However, the pattern of the relationship between Accent and Emotion is the same (or extremely similar) in the three regression curves you present in Figure 1. Perhaps if, in addition to the regression slope, you showed the individual points for each participant, the picture would be easier to read. Additionally, presenting the two regression slopes of the same comparison in the same graph would also make it easier to interpret this effect (that is, at the end, you would have two graphs in Figure 1: one showing the regression slopes for the native and foreign-indo accents, and another one showing the regression slopes for the native and foreign-arab accetns). Same applies to Figure 2.

- I do not understand why you find opposite results in different models: when analyzing Respect and Wrong across Accents (p. 23 and Table 6), you do not find accent effects. However, when you add Emotion to the model (and the interaction terms), you do find simple effects of Accent for Respect and Wrong (NativeVSForeignIndo and NativeVSForeignArab, respectively). These results (and, more concretely, the effect of Accent on Respect for the contrast NativeVsForeignIndo) would also contradict the results of Experiment 1. That makes me think that something failed in these latter models. Please review that the data and models are appropriate.

- The names for the contrasts are different in Tables 9 and 10. Please revise.

- Lines 563-567: this paragraph should be located at the end of the previous section, as it is summarizing the results of Experiments 2, not the results of the cross-experiment analyses. Furthermore, in the latter models you presented, you do find simple effects of Accent on Respect and Wrong. Please revise your analyses and the text, if needed.

- Cross-experimental analyses: I do not follow why you only use the native accent in these analyses. I believe that it would be more interesting to carry out 2-way ANOVAs with Experiment (1 vs. 2) and Accent (native vs. foreign-Indo) for each DV.

- Lines 583-585: But you find that Accent has an effect on Respect/Wrong in Experiment 2 (simple effect), in the model in which you included Emotion and the interaction between Accent and Emotion. As I said before, please revise the data and models, and if the results are still the same, you may change some aspects of the Discussion.

7. PLOS authors have the option to publish the peer review history of their article (what does this mean?). If published, this will include your full peer review and any attached files.

Reviewer #1: No

---

## [Author Response · Author response to Decision Letter 1]

2 Aug 2022

Please, see attach 'Response to Reviewers' document.

---

## [Decision Letter · Decision Letter 2]

6 Sep 2022

Would you respect a norm if it sounds foreign? Foreign-accented speech affects decision-making processes.

PONE-D-22-08900R2

Dear Dr. Foucart,

We’re pleased to inform you that your manuscript has been judged scientifically suitable for publication and will be formally accepted for publication once it meets all outstanding technical requirements.

Kind regards,

José A Hinojosa, Ph.D.

Academic Editor

PLOS ONE

Additional Editor Comments (optional):

Reviewers' comments:

Reviewer's Responses to Questions

**Comments to the Author**

1. If the authors have adequately addressed your comments raised in a previous round of review and you feel that this manuscript is now acceptable for publication, you may indicate that here to bypass the “Comments to the Author” section, enter your conflict of interest statement in the “Confidential to Editor” section, and submit your "Accept" recommendation.

Reviewer #1: All comments have been addressed

2. Is the manuscript technically sound, and do the data support the conclusions?

Reviewer #1: Yes

3. Has the statistical analysis been performed appropriately and rigorously? 

Reviewer #1: Yes

4. Have the authors made all data underlying the findings in their manuscript fully available?

Reviewer #1: Yes

5. Is the manuscript presented in an intelligible fashion and written in standard English?

Reviewer #1: Yes

6. Review Comments to the Author

Reviewer #1: Congratulations to the authors. I think that they were able to improve the quality of their manuscript, and it is now ready to be published.

7. PLOS authors have the option to publish the peer review history of their article (what does this mean?). If published, this will include your full peer review and any attached files.

Reviewer #1: No

---

## [Editor Report · Acceptance letter]

14 Sep 2022

PONE-D-22-08900R2 

Would you respect a norm if it sounds foreign? Foreign-accented speech affects decision-making processes 

Dear Dr. Foucart:

I'm pleased to inform you that your manuscript has been deemed suitable for publication in PLOS ONE. Congratulations! Your manuscript is now with our production department. 

Kind regards, 

on behalf of

Dr. José A Hinojosa 

Academic Editor

PLOS ONE